# Frequency of Hepatitis B Virus Resistance Mutations in Women Using Tenofovir Gel as Pre-Exposure Prophylaxis

**DOI:** 10.3390/v11060569

**Published:** 2019-06-19

**Authors:** Cheryl Baxter, Sinaye Ngcapu, Jason T Blackard, Eleanor A Powell, Patricia K Penton, Salim S Abdool Karim

**Affiliations:** 1Centre for the AIDS Programme of Research in South Africa (CAPRISA), University of KwaZulu-Natal, Private Bag X7, Congella, 4013, Durban, South Africa; sinaye.ngcapu@caprisa.org (S.N.); salim.abdoolkarim@caprisa.org (S.S.A.K.); 2Division of Digestive Diseases, Department of Internal Medicine, College of Medicine, University of Cincinnati, Cincinnati, OH 45267, USA; blackajt@ucmail.uc.edu (J.T.B.); powelleo@ucmail.uc.edu (E.A.P.); pkpenton@gmail.com (P.K.P.); 3Department of Epidemiology, Mailman School of Public Health, Columbia University, New York, NY 10032, USA

**Keywords:** hepatitis B virus (HBV), resistance, tenofovir gel, antiretroviral, pre-exposure prophylaxis

## Abstract

Intermittent use of a single antiretroviral agent in the presence of a replicating virus could potentially increase the development of antiviral resistance. The pericoital, before-and-after sex, dosing regimen used in the Centre for the AIDS Programme of Research in South Africa (CAPRISA) 004 tenofovir gel trial meant that women who were infected with hepatitis B virus (HBV) were exposed intermittently to tenofovir during their participation. The impact of this dosing regimen on HBV resistance was assessed by amplification of the HBV polymerase region from 37 stored plasma samples of women who were HBV surface antigen positive. All samples belonged to HBV genotype A. None of the known tenofovir resistance mutations (M240V/I, L180M, A194T, V214A, N238T) were identified in any individuals. While it is reassuring that no resistance mutations were found among women using topical tenofovir, the rapidly expanding access to oral tenofovir-containing HIV pre-exposure prophylaxis (PrEP), with higher systemic exposure to the drug, makes monitoring for potential HBV drug resistance important.

## 1. Introduction

Although hepatitis B virus (HBV) infection is preventable through effective vaccination, it remains an important cause of morbidity and mortality globally. An estimated 2 billion people alive today have been infected with HBV and more than 257 million are persistent carriers of the disease [1]. Globally, an estimated 80,000 people become infected with HBV each year, and the World Health Organisation estimates that 887,000 people died from HBV-related causes in 2015 [2]. 

Over the past decade, significant advancements have been made with respect to treatment of chronic HBV infection, which has led to a reduction in liver disease and improved outcomes. There are several antiviral agents that are approved for the treatment of HBV, two of which—lamivudine (3TC) and tenofovir disoproxil fumarate (TDF)—are also licensed oral formulations for the treatment of human immunodeficiency virus (HIV). Tenfovir-containing antiretrovirals (as both oral and gel formulations) have been assessed for their ability to prevent new HIV infection [3,4,5], and oral tenofovir-containing pre-exposure prophylaxis (PrEP) is being scaled-up as part of combination HIV prevention strategies in several countries [6,7,8]. 

Given that the same antiretroviral drugs are being used for HIV prevention and HBV treatment, there is a potential for HBV drug resistance developing in people using tenofovir-containing regimens for HIV prevention. Although the incidence of tenofovir resistance following HBV treatment remains low [9,10,11], the development of HBV resistance from PrEP presents a bigger concern as PrEP is often used intermittently. Little is known about the development of HBV resistance with topical or oral prophylactic use of tenofovir.

The CAPRISA 004 trial, which assessed whether topical tenofovir prevents HIV infections [3], included women infected with HBV, and thus provided a unique opportunity to assess the risks and benefits of using tenofovir gel in women with prevalent and incident HBV infection. The intermittent, before-and-after sex dosing regimen used in the CAPRISA 004 trial, where women were only required to use the gel on days that they had sex, meant that women infected with HBV were exposed to variable levels of a single antiretroviral agent during ongoing HBV replication. The purpose of this study was to assess the impact of intermittent topical tenofovir for HIV prevention on the frequency of HBV resistance mutations. 

## 2. Materials and Methods

### 2.1. Study Participants/Samples

The CAPRISA 004 tenofovir gel trial (*n* = 889), conducted between 2007 and 2010, assessed the safety and effectiveness of tenofovir gel for the prevention of HIV [3]. A total of 37 women were infected with HBV (i.e., HBV surface antigen [HBsAg] positive) at enrolment (*n* = 34) or study exit (*n* = 3) [12]. Of these, 20 were assigned to the tenofovir gel group, and 17 were assigned to the placebo group. All 37 women were included in this sub-study, and the screening procedures, informed consent documents and laboratory tests [12] for the HBV safety sub-study were integrated into the CAPRISA 004 trial, including consent for HBV testing and consent for storage of specimens. Stored whole blood plasma samples were used for this analysis. Approval to use the stored samples for this safety sub-study was obtained from the University of KwaZulu-Natal’s Biomedical Research Ethics Committee (BE025/010), 12 February 2010.

### 2.2. Isolation and Amplification of HBV DNA

HBV DNA was extracted from plasma stored at −80 °C from all 37 women. HBV DNA was extracted using the QIAamp DNA blood kit (QIAGEN Inc., Valencia, CA, USA) according to the manufacturer’s instructions. A 1,071 base pair region of the P gene was amplified by direct PCR using the following primers: Werle Pol1-S [5’-TTY CCT GCT GGT GGC TCC AGT TC-3’] and Werle Pol2-AS [5’-CGT CAG CAA ACA CTT GGC-3’] (Integrated DNA Technologies, Coralville, IA, USA). Positive samples were gel purified with the QIAX II Gel Extraction Kit (QIAGEN Inc., Valencia, CA, USA), and seven were cloned into the pGEM-T Easy vector for additional characterization of the P gene. EcoRI restriction digests were carried out using the Miniprep product treated with EcoRI and REact^®^ 3 10× buffer (Life Technologies, Grand Island, NY, USA), and plasmid DNA was purified using the QIAprep Spin Miniprep Kit, according to the supplier’s specifications (QIAGEN Inc., Valencia, CA, USA).

### 2.3. Sequencing of the HBV Pol Region

To assess whether the HBsAg positive women had any resistance mutations and specifically, tenofovir-related resistance mutations, PCR products corresponding to the P gene were sequenced using internal primers and pre-mixed BigDye® Terminator v3.1 Cycle Sequencing kit (Applied Biosystems, Foster City, CA, USA). Sequencing was performed in an automated Applied Biosystems 3130× automated sequencer (Applied Biosystems, Foster City, CA, USA). Each sequence was aligned using Muscle (http://www.drive5.com/muscle/) [13]. Consensus sequence files were BLASTed against the NCBI sequence database in order to identify closely related sequences for downstream phylogenetic analyses. These references were further supplemented with three randomly chosen sequences for each subtype and sub-subtype of HBV (A–F). References were aligned against the sequences from this study in ClustalW (http://www.clustal.org) [14]. The alignment was manually inspected in Geneious software suite (https://www.geneious.com/) and edited until a good codon alignment was achieved. A model test was performed on the edited alignment in jmodeltest2 (https://github.com/ddarriba/jmodeltest2), to estimate the most optimal model of nucleotide substitution for our data. A maximum likelihood (ML) tree topology was constructed from the codon alignment in RaXML v 8.2 [15] with the General Time Reversible model of nucleotide substitution (GTR+G) and 1000 bootstrap replicates [16]. The reference tree and bootstrap trees were analyzed in Booster (https://booster.pasteur.fr/) [17] to infer transfer boostrap support values for branches in the ML-tree topology. The resulting tree topology was visualized, and branches were manually annotated in FigTree (http://tree.bio.ed.ac.uk/software/figtree/). To confirm PCR results, a commercially available drug resistance kit (INNO-LiPA HBV DR v2 and V3 kits (Innogenetics, N.V., Gent, Belgium) that enables DNA material to be hybridised with specific oligonucleotide probes immobilised as parallel lines on membrane-based strips was used.

### 2.4. Identification of Tenofovir-Resistance Mutations

Amino acids 76 to 280 for each sequence were evaluated by their degree of codon polymorphisms and compared with the subtype A reference strain (X02763) to identify mutations in the HBV RT domain. The region analyzed included the five functional domains (A [aa75–91], B [aa163–189], C [aa200–210], D [aa230–241], and E [aa247–257]) and the four interdomains (A–B, B–C, C–D, and D–E). Amino acids for each sequence were evaluated by their degree of codon polymorphisms and compared with the genotype reference strain to identify naturally occurring mutations. The Fisher’s exact test was used to compare the frequency of mutations in the reverse transcriptase (rt) polymerase region in isolates from women in the tenofovir and placebo arms. Variability was classified as follows: conserved regions were defined as site showing variation ≤1%, low polymorphism (5–10% variability), intermediate polymorphism (15–30% variability), and high polymorphism (>50% variability).

## 3. Results

The mean age of the women included in the study was 24.7 years (standard deviation: 5.73) and the median HBV DNA level was 2.63 copies/mL. There were no differences in the clinical and virological characteristics of the 37 women assigned to the tenofovir and placebo groups (Table 1, Appendix A). HBV viral sequences were successfully amplified from 19 of 37 (51%) women including 13 women assigned to tenofovir, and six from women assigned to placebo. In addition, seven (all from women assigned to the tenofovir arm) were cloned into the pGEM-T Easy vector and sequenced, providing a total of 26 HBV sequences assessed for tenofovir resistance mutations. All sequences clustered phylogenetically with HBV genotype A, with the majority (24/26) being subgenotype A1 and two of the isolates clustering with subgenotype A2 (Figure 1).

The C domain, including the YMDD motif, was highly conserved in both the tenofovir and placebo arms (Figure 2). Analysis of the other functional domains revealed the presence of five different amino acid substitutions (with unknown impact on drug resistance) at positions N76, I163, P237, N238 and V253 in women assigned to the tenofovir arm (20 sequences from 13 women) and three different amino acid substitutions at positions N76, P237 and V253 in women assigned to the placebo arm (*n* = 6). Amino acid substitutions in the interdomains occurred at 20 positions in the sequences from women assigned to the tenofovir gel arm and 13 positions in the the sequences from women assigned to the placebo gel arm. The A-B interdomain was highly polymorphic; however, the B-C and D-E interdomains were highly conserved (Figure 2). These finding suggest that any amino acid variation at these highly-conserved sites may have functional constrains on the virus in vivo.

### Prevalence of Polymerase Mutations in Tenofovir-Experienced Versus Tenofovir-Naïve Hbv Sequences

Overall, there were no significant differences in the frequency of amino acid substitutions between the tenofovir-experienced and tenofovir-naïve isolates (all p > 0.05) (Figure 3). A similar number of highly conserved sites (≤1% variation) was found between the isolates from women assigned to the tenofovir 181/204 (88.7%) and placebo arms 188/204 (92.2%). Two (0.8%) of the 204 amino acids in the HBVrt domain were highly polymorphic (≥50% variability)—positions P109 and E125—in both arms. In addition, positions S105, H122 and H271 were highly polymorphic (≥50% variability) in isolates from women assigned to the placebo arm (Figure 3).

Positions N76, R110, S117, I121, T128, L129, R138, L140, V142, W153, I163, L217, P237, N238, R242, T259, K270, V278 and R280 in the isolates from women assigned to tenofovir had ≥5–10% variation, while positions S105, V253, and V266 had ≥15%–30% variation and positions H122 and H271 had >35–40% variation. In isolates from women assigned to the placebo arm, positions N76, T126, T128, W153, L220, T225, C253 and V266 had ≥15%–30% variation and positions S105, P109, R110, H122, Q125, P237, T259, and H271 had ≥30%–50% variation.

Although the frequency of amino acid substitutions were not significantly different, a total of 16 amino acid substitutions in the polymerase domain occurred only in tenofovir-experienced isolates compared to tenofovir-naïve isolates, including S117Y, S117C, I121N, T128I, M129L, R138K, R138E, L140P, V142A, I163V, L217R, N238H, R242K, K270R, V278I, and R280S. Positions R110G, Y126H, T128S, W153R, L220I, T225I, P237H, T259S and H271C were highly polymorphic in tenofovir-naïve compared to tenofovir-experienced isolates. No mutations known to cause tenofovir resistance (L180M, A181I/V, A194T, M204V/I, V214A, Q215S, N236T) or lamuvudine (3TC) resistance (L80V/I, I169T, V173L, L180M, A181T, T184S, M204V/I/S, Q215S) were observed. Absence of these mutations was also confirmed using the INNO-LiPA HBV DR v3 commercial kit (Innogenetics, N.V., Gent, Belgium) (Figure 4).

## 4. Discussion

There was no evidence of HBV drug resistance in women assigned to use tenofovir gel intermittently for HIV prevention in the CAPRISA 004 trial. The absence of well-characterised antiviral resistance mutations in chronically HBV infected individuals in this study is reassuring and may be due to low tenofovir exposure as systemic absorption following gel administration is substantially lower than oral administration [18]. Our findings suggest that intermittent low exposure to tenofovir carries minimal risk of resistance similar to rare emergence of drug resistance from high systemic tenofovir exposure during treatment. Long-term follow-up of patients receiving tenofovir-containing regimens shows that, even after six years of therapy, there were no resistance mutations detected among chronic HBV-infected patients [10,19]. 

Many of the amino acid substitutions observed in this study occurred in both the tenofovir and placebo arms and are unlikely to represent bona fide drug mutations but rather arise naturally during reverse transcription. There were several unique amino acid substitutions in sequences from women assigned to the tenofovir gel arm. Although these less well-characterised amino acid substitutions have an unknown impact on HBV replication fitness, almost all have been previously shown to occur in other genotype A sequences at very low frequencies (<1%) among treatment naïve patients (HBV reference sequences from Genebank). Three of these less well-characterised amino acid substitutions—rtM129L, rtI163V, and rtL217R—are commonly observed in other genotype A sequences, occurring in treatment naïve individuals at frequencies of 44.1%, 5.7% and 36.3%, respectively. The mutation S117C has not been shown to occur in other genotype A sequences but is found in genotypes B, C, and F and among nucleoside reverse transcriptase inhibitor-treated persons. These amino acid substitions need to be further investigated using site-directed mutagenesis to determine their impact on phenotypic resistance, viral replication, and/or polymerase activity. 

Despite these reassuring data, ongoing monitoring of tenofovir-based HBV drug resistance should be incorporated into oral PrEP roll out programmes. As oral PrEP for HIV prevention expands, large numbers of people will likely be exposed to tenofovir in the presence of replicating HBV. Furthermore, data from several clinical trials of oral PrEP have shown that some women, particularly from African settings, find it challenging to adhere to a daily oral PrEP regimen [5,20], thus potentially creating conditions conducive for the development of resistance. Globally, there is a concerning trend of increasing HIV resistance to antiretroviral drugs used for treatment [21], and similar trends in HBV resistance may emerge in the near future as PrEP becomes more widely available. 

This study has several limitations that should be noted. Firstly, the sample size was modest, and HBV DNA was only successfully amplified in half of the women in this study. The low amplification of the samples is possibly due to a low copy number of HBV DNA in the plasma. The median log HBV viral load at study exit was 2.63 copies/mL in the tenofovir gel arm and 1.83 copies/mL in placebo gel arm. Secondly, phenotypic assays were not conducted to establish the relationship between the amino acid substitutions and the development of drug resistance. It was also not possible to establish whether the amino acid substitutions observed were acquired during the trial or if they were pre-existing because stored samples from study exit visits were used to generate the sequences. Pre-existing tenofovir resistance appears to be rare with only one report from a participant in India with a transmitted tenofovir resistance [22]. Nevertheless, the absence of known resistance mutations in blood is reassuring and indicates that vaginally applied tenofovir gel poses little immediate risk of tenofovir resistance emergence and forward transmission of tenofovir-resistant HBV. 

## 5. Conclusions

The findings of this study provide support for the safety of intermittent tenofovir gel use as HIV pre-exposure prophylaxis for HBV-infected individuals. However, the rapidly expanding access to oral tenofovir-containing PrEP for HIV prevention makes the monitoring of potential HBV drug resistance important.

## Figures and Tables

**Figure 1 viruses-11-00569-f001:**
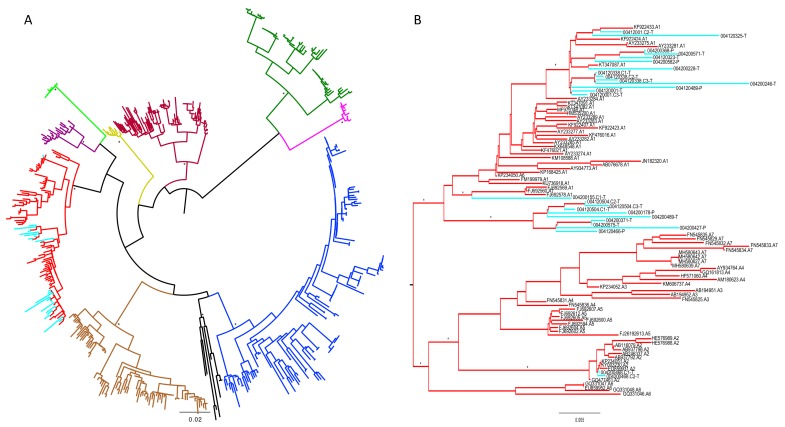
Maximum likelihood tree (midpoint rooted) showing the phylogenetic clustering between South African HBV genome sequences and randomly chosen reference sequences (A–J). The phylogenetic tree was constructed from the codon alignment using RaXML v 8.2 with the General Time Reversible model of nucleotide substitution (GTR+G) and 1000 bootstrap replicates. Tree (**A**) shows all HBV sequences from this study including reference strains, while tree (**B**) shows only clustering between HBV sequences from this study and HBV subtype A reference. Bootstrap support values above 70% are shown with an asterisk (*). HBV sequences from this study are highlighted in light blue, while subtype A references are red. Brown = subtype B reference, blue = subtype C reference, maroon = subtype D reference, yellow = subtype E, dark green = subtype F reference, lumo green = subtype G reference, pink = subtype H reference, purple = subtype I reference, black = subtype J reference.

**Figure 2 viruses-11-00569-f002:**
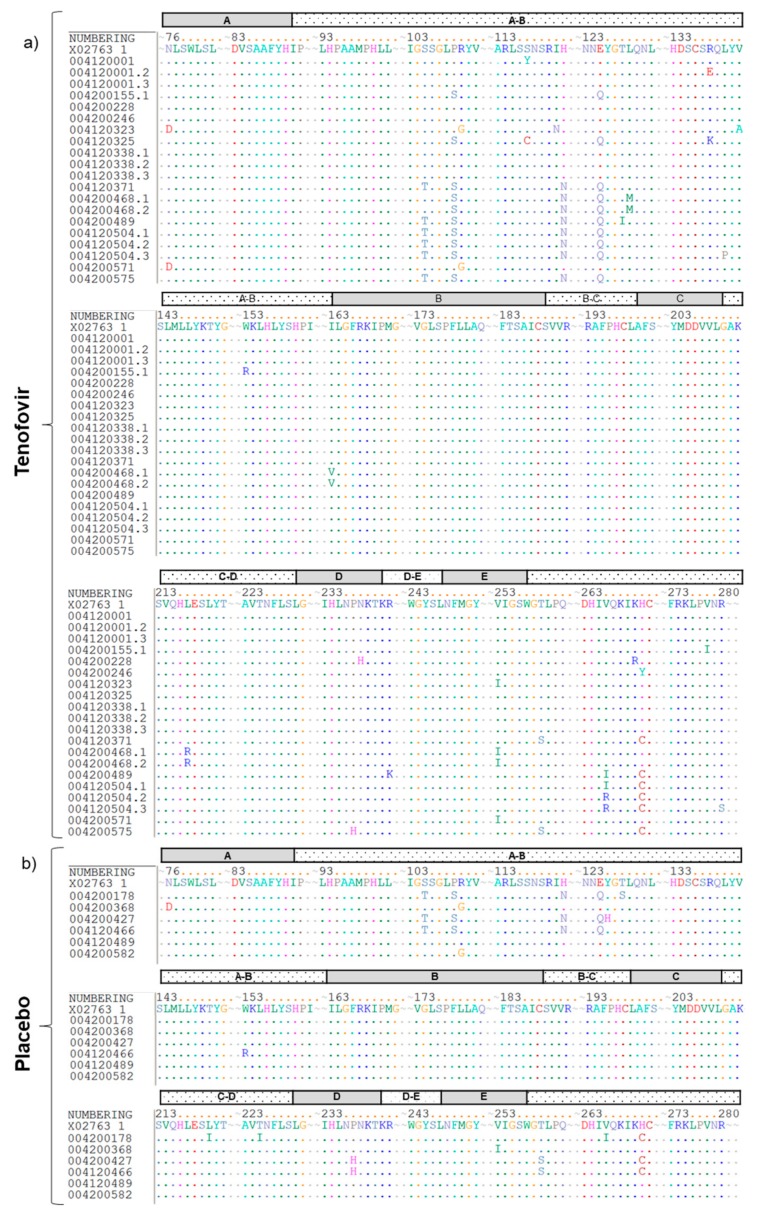
Independent amino acid mutations in the HBVrt domain among women assigned to the (**a**) tenofovir and (**b**) placebo gel arms. GenBank accession number reference sequence X02763 is shown in detail. Each functional domain (A–E) of the HBV RT region is indicated by an alphabetical letter in the shaded box.

**Figure 3 viruses-11-00569-f003:**
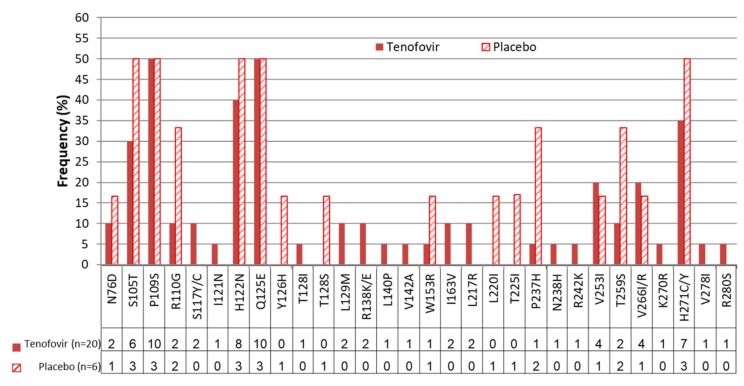
Frequency of the HBVrt domain associated mutations in the tenofovir and placebo isolates.

**Figure 4 viruses-11-00569-f004:**
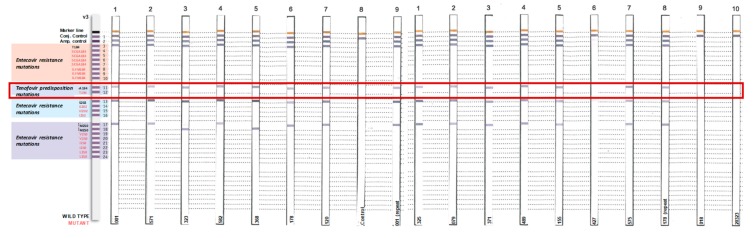
No tenofovir-related mutations were detected among HIV-infected women from the CAPRISA 004 trial with the INNO-LiPA HBV DR v3 strips.

**Table 1 viruses-11-00569-t001:** Clinical and virological characteristics of the 37 women infected with HBV by treatment assignment.

	Overall	Tenofovir (*N* = 20)	Placebo (*N* = 17)	*p* Value
Mean age, years (SD)	24.7 (5.73)	25.15 (5.84)	24.2 (5.74)	0.64
% HBeAg positive (*n/N*)	10.8 (4/37)	10.0 (2/20)	11.8 (2/17)	0.63
Median HBV DNA*Log_10_ copies/mL (IQR)^$^	2.63 (1.1–3.0)	2.61 (1.1–3.2)	1.83 (0.0–3.0)	0.52
Median ALT* (IU/L) (IQR)	20.0 (15.0–22.0)	19.5 (15.8–22.0)	21.0 (15.0–23.0)	0.66
Median AST* (IU/L) (IQR)	22.0 (19.0–25.0)	23.5 (19.8–25.3)	20.0 (19.0–24.0)	0.31
% with any grade 2^ or higher liver related adverse event during follow up (*n/N*)	18.9 (7/37)	10.0 (2/20)	29.4 (5/17)	0.21

* results from study exit samples. $ Two participants had no HBV DNA results due to insufficient storage sample available for testing in 1 participant and 1 participant was lost to follow-up; ^ grade 2 adverse event defined as >2.5 upper limit of normal.

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
