# Peer review of "Frequency of Hepatitis B Virus Resistance Mutations in Women Using Tenofovir Gel as Pre-Exposure Prophylaxis"

_viruses, 2019, doi:10.3390/v11060569_

Round 1

Reviewer 1 Report

This study is aimed at investigating if the use of tenofovir gel in HIV+HBV coinfected women is associated with the emergence of drug-resistance mutations in HBV reverse transcriptase. No drug resistance mutations were analysed. Furthermore, the low number of patients did not allow to define statistically significant differences in the prevalence of amino acid substitutions in the two group of women analysed (TDF arm vs placebo). There are several points to be clarified:

.     

1.     A table reporting the clinical and virological characteristics of the 37 women included in the study should be included in the manuscript. The Table should include levels of serum HBV-DNA, ALT/AST, HBeAg status, quantitative HBsAg (if available). I assume that all women were drug-naïve, please report therapy status.

2.     The rate of successful sequencing is low. Can it be due to low serum HBV-DNA?

3.     Which was the subgenotype observed in HBV infected women?

4.     Please, better explain “greater than 5% variation” or “greater then 17% variation”and why the authors decided to choose this thresholds, and how they have been calculated.

5.     Please, verify the concordance between the prevalence of amino acid substitutions reported in the text and those reported in Figure 2.

6.     Please, explain the meaning of X in some amino acid substitutions are reported.

7.     The authors reported “a total of 22 amino acid substitutions in the polymerase domain were positively selected in tenofovir-experienced isolates compared to tenofovir-naïve isolates”. How was positive selection measured?”

8.     In Figure 3, please report the total number of patients used to calculate the %

9.     What about the impact of TDF gel on the risk of acquiring HBV infection in the cohort of CAPRISA 004 trial? This can be an interesting result to be included.

Author Response

There are several points to be clarified:

1.       A table reporting the clinical and virological characteristics of the 37 women included in the study should be included in the manuscript. The Table should include levels of serum HBV-DNA, ALT/AST, HBeAg status, quantitative HBsAg (if available). I assume that all women were drug-naïve, please report therapy status.

A table reporting the clinical and virological characteristics of the women included in this analysis has been added. All women were drug naïve. A summary table has been added to the results, and a table with the individualised data has been added a supplementary Table 1.

Table 1: Clinical and virological characteristics of the 37 women infected with HBV by treatment assignment

Overall

Tenofovir (n=20)

Placebo (N=17)

P value

Mean age, years (SD)

24.7 (5.73)

25.15 (5.84)

24.2 (5.74)

0.64

% HBeAg positive (n/N)

10.8 (4/37)

10.0 (2/20)

11.8 (2/17)

0.63

Median HBV DNA*Log10   copies/ml (IQR)$

2.63 (1.1 – 3.0)

2.61 (1.1 – 3.2)

1.83 (0.0-3.0)

0.52

Median ALT* (IU/L) (IQR)

20.0 (15.0 - 22.0)

19.5 (15.8 – 22.0)

21.0 (15.0 – 23.0)

0.66

Median AST* (IU/L) (IQR)

22.0 (19.0 - 25.0)

23.5 (19.8 – 25.3)

20.0 (19.0 – 24.0)

0.31

% with any grade 2^ or higher liver   related adverse event during follow up (n/N)

18.9 (7/37)

10.0 (2/20)

29.4 (5/17)

0.21

*results from study exit samples. $Two participants had no HBV DNA results due to insufficient storage sample available for testing in 1 participant and 1 participant was lost to follow-up; ^grade 2 adverse event defined as >2.5 upper limit of normal

Supplementary Table 1: Clinical and virological characteristics of the 37 women infected with HBV

PID

Age (Years)

Group assignment

HBeAg status

HBV DNA (copies/ml)*

ALT* (IU/L)

AST* (IU/L)

Any grade 2^ or higher liver related adverse   event during follow up

120001

25

Tenofovir

Positive

5.61

48

50

yes

120018

20

Tenofovir

Negative

1.08

11

19

no

120078

18

Tenofovir

Negative

1.08

19

17

no

120176

24

Placebo

Negative

1.08

17

20

yes

120205

20

Placebo

Negative

0.00

39

30

yes

120253

37

Tenofovir

Negative

1.08

20

21

no

120260

29

Placebo

Negative

2.93

23

19

no

120314

37

Tenofovir

Negative

2.81

22

24

no

120323

27

Tenofovir

Negative

3.02

15

19

no

120325

30

Tenofovir

Negative

4.78

26

27

no

120338

22

Tenofovir

Negative

2.30

16

21

no

120466

26

Placebo

Negative

3.03

14

21

no

120489

20

Placebo

Negative

0.00

15

19

no

120499

18

Placebo

Negative

3.62

22

22

yes

120504

29

Tenofovir

Negative

2.63

22

24

no

145008

24

Tenofovir

Negative

2.59

24

25

no

145016

39

Placebo

Negative

$

18

18

no

200079

27

Placebo

Negative

1.56

31

31

no

200120

19

Placebo

Negative

2.67

22

22

no

200125

24

Tenofovir

Positive

0.00

20

36

no

200155

22

Tenofovir

Negative

1.95

22

26

no

200178

31

Placebo

Negative

1.67

15

18

no

200228

23

Tenofovir

Negative

3.16

10

25

no

200246

18

Tenofovir

Negative

2.16

31

25

no

200323

22

Placebo

Positive

0.00

12

15

yes

200342

24

Placebo

Negative

0.00

18

17

no

200355

32

Tenofovir

Negative

0.00

8

18

no

200368

18

Placebo

Positive

8.04

33

56

no

200371

30

Tenofovir

Negative

3.54

22

20

no

200427

31

Placebo

Negative

2.91

22

28

no

200468

26

Tenofovir

Negative

$

12#

17#

no

200489

22

Tenofovir

Negative

3.97

18

22

no

200515

19

Placebo

Negative

2.99

21

19

no

200533

24

Placebo

Negative

1.83

14

19

yes

200571

19

Tenofovir

Negative

2.72

18

26

no

200575

18

Tenofovir

Negative

3.21

19

23

yes

200582

21

Placebo

Negative

3.71

26

24

no

*results from study exit samples. $insufficient storage sample available for testing in 1 participant and 1 participant was lost to follow-up; ^grade 2 adverse event defined as >2.5 upper limit of normal; #results from study month 11

2.       The rate of successful sequencing is low. Can it be due to low serum HBV-DNA?

We have included some speculation in the limitations section of the discussion to explain the low rates of successful sequencing in this study as follows:

“The low amplification of the samples is possibly due to a low copy number of HBV DNA in the plasma. The median log HBV viral load at study exit was 2.63 copies/mL in the tenofovir gel arm and 2.25 copies/mL in placebo gel arm.”

In addition, the table of clinical and virological characteristics includes the HBV DNA levels.

3.       Which was the subgenotype observed in HBV infected women?

We have added the subgenotypes as follows:

“All sequences clustered phylogenetically with HBV genotype A, with the majority (24/26) being subgenotype A1 and two isolates clustering with subgenotype A2 (Figure 1).”

4.       Please, better explain “greater than 5% variation” or “greater then 17% variation” and why the authors decided to choose this thresholds, and how they have been calculated.

The greater than 5% variation and greater than 17% variation represented amino acid substitutions occurring in more than 1 sequence in the tenofovir group (1/20, i.e. 5%) and in more than 1 sequence placebo group (1/6, i.e. 17%). These phrases have been removed to avoid confusion.  

In addition, the methods section has been expanded to clarify the classification of variability used in this study as follows:

“Amino acids for each sequence were evaluated by their degree of codon polymorphisms and compared with the genotype reference strain to identify naturally occurring mutations. The Fisher’s exact test was used to compare the frequency of mutations in the reverse transcriptase (rt) polymerase region in isolates from women in the tenofovir and placebo arms. Variability was classified as follows: conserved regions were defined as site showing variation £1%, low polymorphism (5 - 10% variability), intermediate polymorphism (15 – 30% variability), and high polymorphism (>50% variability).”

5.       Please, verify the concordance between the prevalence of amino acid substitutions reported in the text and those reported in Figure 2.

The concordance between the prevalence of amino acid substitutions reported in the text and those in Figure 2 have been verified as suggested.

6.       Please, explain the meaning of X in some amino acid substitutions are reported.

Substitutions named “X” are sequencing ambiguities. There were four instances where the “X” appeared, three of which were from a single sequence (004120371). We have rechecked the sequences and removed the “X” and replaced the appropriate amino acid where applicable. The revised Figure 2 has been included.

7.       The authors reported “a total of 22 amino acid substitutions in the polymerase domain were positively selected in tenofovir-experienced isolates compared to tenofovir-naïve isolates”. How was positive selection measured?”

The sentence has been edited to more clearly indicated that these substitutions occurred only in the tenofovir group. In addition, the number of amino acid substitutions occurring only in the tenofovir arm has been updated following the checking of concordance between the text and figure 2.

“….a total of 16 amino acid substitutions in the polymerase domain were positively selected occurred only in tenofovir-experienced isolates compared to tenofovir -naïve isolates,…”

8.       In Figure 3, please report the total number of patients used to calculate the %

A data table has been added to figure 3 to show the number of patients used to calculate the percentage

9.       What about the impact of TDF gel on the risk of acquiring HBV infection in the cohort of CAPRISA 004 trial? This can be an interesting result to be included.

This particular issue has already been addressed in a previous publication (Baxter, C., N. Yende-Zuma, P. Tshabalala, Q. Abdool Karim and S. S. Abdool Karim (2013). "Safety of coitally administered tenofovir 1% gel, a vaginal microbicide, in chronic hepatitis B virus carriers: results from the CAPRISA 004 trial." Antiviral Res 99(3): 405-408.)

A total of 22 women acquired HBV infection during the 878.7 women-years (mean = 18 months) of follow-up: 14 and 8 in the tenofovir and placebo gel arms, respectively. The HBV incidence rate in the tenofovir 1% gel arm was 3.2 per 100 women-years (wy) (95% CI: 1.7, 5.3) compared to 1.8 per 100 wy (95% CI: 0.8, 3.6) in the placebo gel arm (IRR = 1.8; 95% CI: 0.7, 4.8; p = 0.21).

Reviewer 2 Report

This is an interesting and well written study reassuring that vaginally applied tenofovir gel poses minimal or no risk for tenofovir resistance emergence.

Minor changes

Line 22 (abstract) “subtype A” should be replaced by “genotype A” (same for figure 1 legend)

Line 41 Please correct “Tenfovoir”

Author Response

Minor changes

1.     Line 22 (abstract) “subtype A” should be replaced by “genotype A” (same for figure 1 legend)

The word “subtype” has been replaced with “genotype”

2.     Line 41 Please correct “Tenfovoir”

The spelling of tenofovir has been corrected

Round 2

Reviewer 1 Report

I thank the authors for their satisfactory answers